**Data Availability Statement:** All data and related metadata underlying the findings reported are included in the submitted article and its supplementary information files.

# What are the barriers and facilitators to polio vaccination and eradication programs? A systematic review

Obidimma Ezezika[1,2,3]*, Meron Mengistu[2,4], Eric Opoku[2,5], Aiman Farheen[2], Anuradha Chauhan[2], Kathryn Barrett[1,6]

1 School of Health Studies, Faculty of Health Sciences, University of Western Ontario, London, Canada, 2 Department of Health and Society, University of Toronto Scarborough, Toronto, Canada, 3 African Centre for Innovation and Leadership Development, Abuja, Nigeria, 4 London School of Hygiene & Tropical Medicine, Keppel St, London, United Kingdom, 5 Institute of Health Policy, Management and Evaluation, University of Toronto, Toronto, Canada, 6 University of Toronto Scarborough Library, Toronto, Canada

* oezezika@uwo.ca

## Abstract

Global efforts to eradicate polio by the Global Polio Eradication Initiative agency partners and country-level stakeholders have led to the implementation of global polio vaccination programs. This study presents the findings of existing studies regarding the barriers and facilitators that countries face when implementing polio interventions. A comprehensive search was conducted in OVID Medline, OVID Embase, EBSCO CINAHL Plus, and Web of Science. Eligible studies underwent quality assessment. A qualitative evidence synthesis approach was conducted and aligned to the Consolidated Framework for Implementation Research (CFIR). The search identified 4147 citations, and following the removal of duplicates and screening according to our inclusion/exclusion criteria, 20 articles were eligible for inclusion in the review. Twelve countries were represented in this review, with India, Nigeria, Pakistan, Ethiopia, and Afghanistan having the most representation of available studies. We identified 36 barriers and 16 facilitators. Seven themes emerged from these barriers and facilitators: fear, community trust, infrastructure, beliefs about the intervention, influential opinions, intervention design, and geo-politics. The most frequently cited CFIR constructs for the facilitators and barriers were *knowledge and beliefs about the intervention*, followed by *available resources*. This study identified a wide range of barriers and facilitators to polio vaccination implementation across the globe, adding to the scarce body of literature on these barriers and facilitators from an implementation perspective and using a determinant framework. The diversity of factors among different groups of people or countries highlights the relevance of contexts. Implementers should be conversant with the contexts within which polio eradication programs boost intervention coverage and capacity. This study provides policymakers, practitioners, and researchers with a tool for planning and designing polio immunization programs.

**Trial registration:** A protocol for this systematic review was developed and uploaded onto the PROSPERO international prospective register of systematic reviews database (Registration number: CRD42020222115).

**Funding:** The authors received no specific funding for this work.

**Competing interests:** The authors have declared that no competing interests exist.

## Introduction

The Polio Eradication Strategy 2022 to 2026 by the Global Polio Eradication Initiative (GPEI) agency partners aims at surmounting all implementation barriers and optimizing existing strategies that have enabled progress over the years [1]. An important channel for achieving this objective is exploring existing and emerging barriers to and facilitators of polio eradication programs to guide future research and policy implementation strategies globally.

The implementation of polio vaccination and eradication programs has met varying success from the perspectives of countries, regions, or stakeholders, and it can be synthesized to give a broader overview of issues. Specifically, evidence synthesis of the barriers to and facilitators of polio eradication programs improves complex health interventions [2], especially in resource-constrained settings. Numerous researchers have documented implementation factors with regard to polio vaccination and eradication in the literature [2–7]. Systematically reviewing these lessons across nations may be relevant for the formulation and implementation of other vaccination programs.

Over the last five years, there have been at least two systematic reviews related to polio eradication and vaccination [2, 8]. Alonge et al. used surveys and key informant interviews in seven countries (i.e., Ethiopia, Nigeria, Afghanistan, Indonesia, India, Bangladesh, and the Democratic Republic of Congo) to synthesize knowledge from polio eradication initiatives and the internal and external contextual factors that may serve as barriers or facilitators to these initiatives [9]. The study results were reported based on the consolidated framework for implementation research (CFIR). The researchers found that external factors (e.g., social, political, and economic) were the most cited barriers to polio eradication activities. Moreover, they did not find significant facilitators to polio eradication.

Currently, the relevance of employing qualitative evidence synthesis (QES), a methodology that identifies factors from qualitative research studies, has been noted in the literature as a way to improve complex health interventions, such as polio eradication [2]. Although not conducted from a global perspective, several systematic reviews have yielded insights to the factors around polio eradication. For example, using QES and the best-fit framework, Mshelia et al. conducted a systematic review of factors that influence the implementation of the GPEI in low- and middle-income countries [2]. The study highlighted the relevance of a robust supporting infrastructure, such as well-trained staff in polio eradication initiatives. Ataullahjan et al. conducted a systematic review of the barriers and facilitators of polio eradication initiatives in Pakistan [8]. The results were divided into caregivers' beliefs and experiences, the Pakistani Polio program, and threats to Pakistan's polio eradication initiatives.

Polio vaccination and eradication interventions are complex, so the relevance of a global and broader perspective cannot be overemphasized.

This study intends to present its findings from the global context using the QES approach. The results of this study may guide researchers and decision- and policy-makers on the important barriers and facilitators of polio interventions to be explored.

## Methods

This systematic review is presented according to the preferred reporting items for systematic reviews and meta-analyses (PRISMA) 2020, found in S1 Data [10]. A protocol for this study is registered in the international prospective register of systematic reviews, PROSPERO CRD42020222115 [11].

### Search strategy

A comprehensive search was conducted in OVID Medline, OVID Embase, EBSCO CINAHL Plus, and Web of Science according to a search strategy developed by an academic health

**Table 1. Summary of inclusion and exclusion criteria.**

| Selection Criteria | Inclusion Criteria | Exclusion Criteria |
|---|---|---|
| | *Publication Characteristics* | |
| **Language** | English | All languages except English |
| **Publication Type** | Scholarly journal articles | All publications that are not scholarly journal articles |
| **Study Type** | Primary (research) | Secondary (review) |
| **Study Methods** | Qualitative or mixed methods with a qualitative component | Quantitative |
| | *Study Characteristics* | |
| **Issue** | Eradication of polio or polio vaccination | All health issues except the eradication of polio or polio vaccination |
| **Intervention** | Intervention, program or campaign to eradicate polio, including polio vaccination | No discussion of an intervention, program, or campaign to eradicate polio |
| **Outcome** | At least one barrier or facilitator to the implementation of the intervention, program, or campaign | No discussion of the barriers and facilitators to implementation of the intervention, program or campaign |

sciences librarian. The search was executed on September 18, 2020. The results were limited to English-language journal articles involving human subjects. No publication date limits were applied. The search strings used in each database are provided in S2 Data.

## Eligibility criteria

To be included in the review, the studies needed to be English-language, primary research articles published in academic journals. The research needed to employ qualitative methods or mixed methods with a qualitative component. Studies needed to report on the implementation of a polio eradication or polio vaccination intervention, program, or campaign and address at least one barrier or facilitator related to the implementation process. The intervention could take place globally in any setting and involve participants of any age. The eligibility criteria are presented in Table 1.

## Study selection

Study screening was completed using Covidence, a systematic review management tool, through a two-step process (Fig 1). After pilot testing the project's screening guidelines document to ensure consensus between team members, all article titles and abstracts were screened independently by two reviewers. Conflicts were resolved by a third party. Next, the full texts of each study were retrieved and imported into Covidence. Three reviewers were involved in full-text screening, with each article being screened independently by two individuals. During full-text screening, the screeners focused on ensuring that studies were about implementation rather than adoption because the screeners had been over-inclusive during title and abstract screening on this criterion, and we wanted to ensure that the eligibility criteria were strictly followed. Following group discussion, a third party resolved conflicts between the reviewers.

## Data extraction

After pilot testing the data extraction template, two reviewers independently performed data extraction for the included articles. The data extraction template captured the following items: study title, author, year of publication, country, methodology, participants, study setting, study objectives, and barriers and facilitators to implementation of the intervention. Conflicts in data extraction were resolved by a third party. For the barriers and facilitators, conflicts could

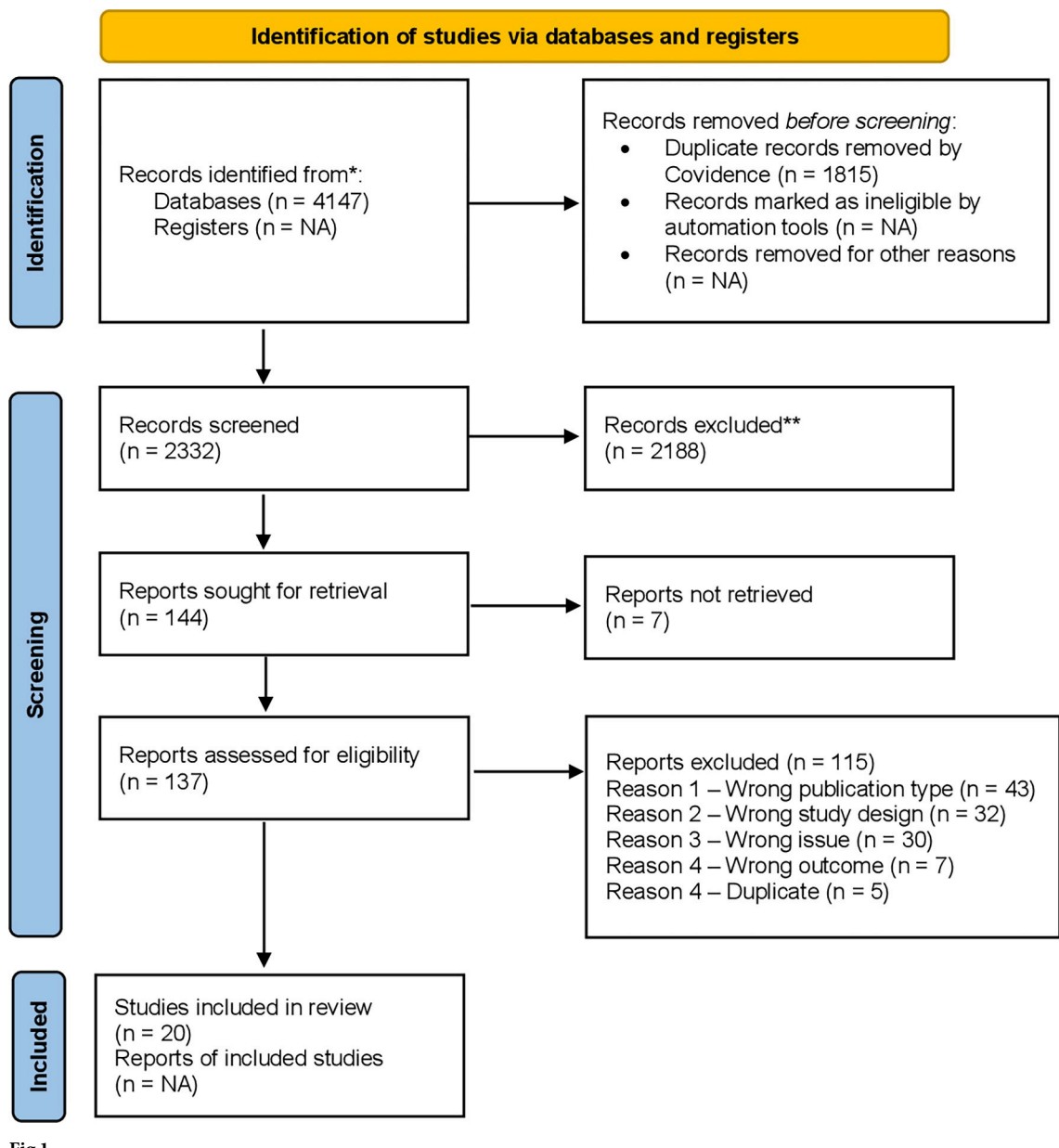

**Fig 1.**

include cases where one reviewer noted a barrier or facilitator while the other did not, as well as differences in naming the barrier or facilitator.

Descriptive information about each study was compiled in a study characteristics table. All data related to barriers and facilitators were compiled in a consensus document, which included the name of every barrier and facilitator extracted from each study, the sentence in which the barrier or facilitator was mentioned, and a longer excerpt from the study showing the barrier or facilitator in context. The consensus document was used as the basis for collapsed tables of the barriers and facilitators, in which one team member grouped the same barriers and facilitators together and listed all the studies in which these barriers or facilitators were identified (including the passage mentioning the barrier/facilitator and its location in the full-text document). Two other team members reviewed and ultimately approved the collapsed tables.

### Data synthesis

Two team members used the CFIR framework as a guide to organize the barriers and facilitators identified in the data extraction process. To ensure a high level of agreement, the reviewers performed a pilot test on the first two barriers and facilitators and, following a consensus meeting to review their work, coded the first 20 barriers. At a subsequent consensus meeting, the consensus level was revealed to be low due to inconsistencies in barrier titles and supportive quotes. To ensure the coding was accurate, the two team members reviewed all barrier and facilitator titles and quotes together, and a third party confirmed the accuracy of the changes made by the researchers.

The two team members then independently coded the rest of the barriers and facilitators, which were reviewed by a third party for accuracy. Those that could not be coded with the CFIR were inductively coded. The final consensus meeting for all barriers and facilitators was conducted with all three parties to produce the results in S1 and S2 Tables. All the barriers and facilitators were inductively grouped into broader themes.

### Quality assessment

The confidence level of the review findings was assessed through inter-rater reliability scores for both barriers and facilitators. The researchers had inter-rater reliability scores of 89 and 80 percent for the barriers and facilitators, respectively. Discrepancies were resolved through discussions with a third party who served as the final decision-maker. The Mixed-Methods Appraisal Tool (MMAT) and Critical Appraisal Skills Programme (CASP) were used to assess the quality of the articles [12, 13]. Both researchers independently reviewed the primary research studies and categorized them using the appraisal tool. Discrepancies between researchers' results were resolved by a third party.

## Results

The studies included in this systematic review were analyzed to extract study characteristics, including study methods, years of publication, participants, countries, and study objectives. Details are included in S3 Table.

### Type of study/methods

The articles included in our systematic review used two kinds of designs, including qualitative and mixed methods approaches. Details are in Table 2.

The most common study design utilized across all studies was a qualitative approach (n = 12). Data collection methods across all qualitative studies included focus group discussions (n = 5), interviews (n = 8), participant (and non-participant) observation (n = 2/n = 2), and document/report/literature reviews (n = 3).

The second most frequently used study design across all studies was a mixed methods approach (n = 8). The qualitative data collection methods were like those studies exclusively using a qualitative approach, including interviews (n = 6), focus group discussions (n = 1), and participant observation (n = 1). Cross-sectional surveys were an additional data collection method (n = 2). Quantitative data collection methods in studies utilizing a mixed methods approach included interviews (n = 2), surveys (n = 2), and national resources (including national records and reports) (n = 3).

### Year of publication

The included articles were published as early as 1960 and as recently as 2019. Two articles were published between 1960–1970. Between 1970–1999, there were no published articles included

**Table 2. Frequency table of study-identified barriers* and facilitators (n = 20).**

| CFIR domains (n = 5) and constructs (n = 39) | Barrier n (%) of studies | Facilitator n (%) of studies |
|---|---|---|
| **I. Intervention characteristics** | | |
| *No facilitators or barriers were noted for the following constructs: Intervention source, relative advantage, adaptability, trialability, complexity, and cost* | | |
| Evidence Strength and Quality | 1 (5%) | None identified |
| Design Quality and Packaging | 3 (15%) | 1 (5%) |
| **II. Outer Setting** | | |
| *No facilitators or barriers were notes for the following constructs: Cosmopolitanism, and Peer pressure* | | |
| Patient Needs and Resources | 1 (5%) | None identified |
| External Policy and Incentives | None identified | 1 (5%) |
| **III. Inner Setting** | | |
| *No facilitators or barriers were notes for the following constructs: Structural characteristics, Networks and communications, Culture, Implementation climate, Relative priority, Organizational incentives and rewards, Goals and feedback, Learning climate, Readiness for implementation, Leadership engagement, access to knowledge and information* | | |
| Tension for Change | None identified | 2 (10%) |
| Compatibility | 2 (10%) | None identified |
| Available Resources | 9 (45%) | 4 (20%) |
| **IV. Characteristics of Individuals** | | |
| *No facilitators or barriers were notes for the following constructs: Self-efficacy, Individual stage of change, and Individual identification with organization* | | |
| Knowledge and Beliefs about the Intervention | 17 (85%) | 8 (40%) |
| Other Personal Attributes | 4 (20%) | None identified |
| **V. Process** | | |
| *No facilitators or barriers were notes for the following constructs: Planning, Engaging, Formally appointed internal implementation leaders, Executing, and Reflecting and Evaluating* | | |
| Opinion Leaders | 1 (5%) | None identified |
| Champions | None identified | 4 (20%) |
| External Change Agents | 10 (9%) | 1 (5%) |

* Three barriers that could not be coded with the CFIR. These include: the relocation of mothers, the inaccessibility to reach children, and conflict and security issues.

in our study. Between 2000–2010, three articles were published and included in our review. The bulk of the articles used in our study were published between 2010–2020 (n = 14).

## Participants

The most frequently used target population across all studies in our review included mothers and caregivers of vaccine-eligible children (n = 14). Other participants included those in implementation, distribution, and community roles. For instance, those in health representative roles, including health officials, political leaders, the World Health Organization, and other international organizations, as well as other decision-making and implementing stakeholders, were recruited as participants across several studies (n = 7). Moreover, health providers, including community health workers, immunization service providers, polio program staff, and other ground-level staff, were included across various studies (n = 8). Religious leaders were notable participants in a few studies, as their opinions were often cited as influential in caregivers' decisions around polio vaccine uptake (n = 2). Other participants included the general population through household survey participation (n = 4), community members (n = 3), and health journalists (n = 1).

## Countries

A total of 12 countries were included across all studies, including Liberia, India, Ethiopia, Nigeria, Rwanda, Angola, Nepal, Pakistan, the United Stated of America, Nigeria, Niger, and Uganda. Across continents, the most frequently cited setting of study was South Asia. The most frequently studied country was Pakistan (n = 8), with India following (n = 6). Three studies adopted a global analysis, including multiple countries across several continents, as their settings.

## MMAT/CASP

The MMAT and the CASP Qualitative Research Checklist were used to assess the quality of the studies included in the review that used mixed methods and qualitative study designs, respectively. The MMAT and CASP tools were used to categorize and identify methodological quality through a set of criteria. S3 Data provides the MMAT/CASP results for all 20 studies.

Of the 20 studies included in the review, 12 adopted qualitative study designs. All 12 utilized appropriate research designs to address the aims of their research and had appropriate recruitment strategies to investigate the aims of the research. Regarding the relationship between research and participants, it was unclear in two studies whether this relationship was considered, and in two other studies, this was not considered. In addition, all qualitative studies were deemed to produce valuable research through their methodological strategies.

Eight studies utilized a mixed methods approach. Adequate rationale for using a mixed methods design was addressed by all eight studies to explore their research questions. Seven studies had outputs of the integration of qualitative and quantitative components that were adequately interpreted in their studies, whereas one study did not. The different components of all eight studies adhered to the quality criteria of each tradition of the methods involved.

## Study objectives of the selected papers

Across all 20 articles, several objectives emerged. Seven studies stated that their primary objective was to investigate public knowledge, perceptions, and distrust, which influenced acceptance of the polio vaccine across several settings. In addition, the investigation of the impact of polio eradication activities on routine immunization and primary healthcare were cited as objectives across five studies. Several other objectives were highlighted as well. For instance, analyzing the challenges that the GPEI encountered and the approaches taken to address these challenges was the objective of two studies. Across the mixed methods studies, uncovering the association between social characteristics and immunization coverage was a cited objective (n = 2).

Other study objectives included investigating social and cultural factors that affect detection and reporting of disease cases in a surveillance system (n = 2), investigating factors underlying changes in vaccine coverage (n = 1), analyzing the challenges that routine immunization programs faced post-Ebola (n = 1), and investigating factors influencing attendance for polio National Immunization Days (n = 1).

## Barriers and facilitators

The researchers identified a total of 36 barriers and 16 facilitators across all 20 studies. The CFIR assisted researchers in organizing and grouping the barriers and facilitators identified into constructs (Table 2). The researchers identified seven overarching themes from the analysis of the results, including fear of vaccine outcomes, community trust, infrastructure, beliefs about the intervention, influential opinions, intervention design, and geo-politics (Table 3).

**Table 3. Themes of barriers and facilitators.**

| Themes | Description | Barriers | Facilitators |
|---|---|---|---|
| **Fear** | *The fear associated with the vaccine including its purpose, the motives of the stakeholders involved, and reproductive side effects. It also includes the fear associated with contracting polio.* | ❍ Presence of rumors (1) <br> ❍ Suspicions of immunization side effects (7) <br> ❍ Fear of vaccination being experimental (1) <br> ❍ Fear of vaccines due to past Ebola experiences (1) <br> ❍ Fear of illness of children/parents (3) | ❍ Fear of contracting the disease (1) |
| **Community trust** | *The level of trust in organizations and government systems and how this affects vaccination uptake in communities.* | ❍ Government distrust (1) <br> ❍ Resistance due to lack of basic health facilities (1) <br> ❍ Refusal as a means of protest (1) | ❍ Increased trust in government health services (1) |
| **Infrastructure** | *The general infrastructure needed to support polio immunization programs, including health post and information access, community reach, organization, and quality.* | ❍ Lack of basic necessities (1) <br> ❍ Lack of access to healthcare (1) <br> ❍ Poor financial, organizational, logistical, and OPV quality constraints (2) <br> ❍ Unavailability of children (4) <br> ❍ Lack of infrastructure (1) <br> ❍ Inconvenience to vaccine access (1) | ❍ Good social mobilization system (1) <br> ❍ Finding and mapping marginalized population (1) <br> ❍ Systematic identification and vaccination of mobile and migratory populations (1) <br> ❍ Improvements in infrastructure (1) <br> ❍ Increased accessibility to immunization due to close NID posts (1) |
| **Beliefs about the intervention** | *Beliefs about the intervention associated with cultural and religious conceptions, and the level of knowledge and awareness about the vaccine's purpose and effectiveness.* | ❍ Anti-immunization debates <br> ❍ Propaganda and mistrust against vaccines and its <br> ❍ efficacy (3) <br> ❍ Cultural beliefs (1) <br> ❍ Lack of information, ignorance and illiteracy (8) <br> ❍ Belief that vaccination was <br> ❍ unnecessary (4) <br> ❍ Vaccine-resistant areas (1) <br> ❍ Safety and religious misconceptions regarding vaccine (2) <br> ❍ Participant perception of advanced age (1) <br> ❍ Lax or disinterest of parents concerning vaccinations (1) <br> ❍ Reliance on religious leaders that oppose OPV (1) <br> ❍ Mothers avoiding OPV (1) | ❍ Parental moral obligation and inability to be deviant (1) <br> ❍ Belief in the vaccine's ability to protect against polio (4) <br> ❍ Knowledge/awareness of polio vaccines (3) <br> ❍ Decrease in religious resistance to OPV (1) |
| **Influential opinions** | *The association between the opinions of health professionals and religious leaders, and the acceptance of vaccination. This includes religious leaders and health professionals serving as influential voices in the decision-making of parents and communities.* | ❍ Medical practitioners' opposition against repeated rounds of polio (1) <br> ❍ Health worker and opinion leaders' recommendations (1) <br> ❍ Protests by alternative medicine providers (1) <br> ❍ Reliance on religious leaders that oppose OPV (1) | ❍ Influence from health-focused champions (3) <br> ❍ Involvement of religious scholars (1) <br> ❍ Decrease in religious resistance to OPV (1) |
| **Intervention design** | *The quality, design presentation, and policies in support of the intervention. This includes coordinating visits of vaccination teams, campaigns, and vaccination mandates.* | ❍ Absence or failure of vaccination teams visiting homes (2) <br> ❍ Frequent visits (1) | ❍ Multiple polio campaigns (1) <br> ❍ Government mandating polio vaccinations to fly abroad (1) <br> ❍ Coercive means (2) |
| **Geo-politics** | *Issues related to geography politics, governance, and security terrain of a location in which the intervention is being implemented.* | ❍ Relocation of mothers (2) <br> ❍ Inaccessibility to reach children (1) <br> ❍ Conflict and security issues (2) | |

*Note.* The number in brackets indicates the number of participants that mentioned each barrier or facilitator.

**Fear.** Fear was a prominent theme in the implementation of polio programs. Fear was associated with rumors, suspicions, and past experiences with other diseases, rendering parents reluctant to take up polio programs. Coincidentally, fear also facilitated program implementation as fear of illness due to polio increased vaccination uptake. Under this theme, nine studies mentioned a total of five barriers, and one study highlighted fear as a facilitator.

One study stated that the presence of rumors discouraged families in Ethiopia from having their children vaccinated due to fears of inducing sterility and/or disability [14]. This same study, along with six others, highlighted suspicions of immunization side effects as a barrier to implementation [14–20]. Suspicions included actual and perceived side effects, ulterior motives of the government, and infertility. For instance, one study described that in India, "most [participants] said that there was resistance among the population in some areas due to the fear that OPV [oral polio vaccine] causes infertility" (p. 5) [17]. Moreover, one study highlighted the fear that families had regarding the vaccination being experimental [21]. Lastly, one study identified fear of vaccines due to past Ebola experiences as a deterrent to polio vaccine uptake [22]. Attributing fears related to past Ebola experiences in the region, participants in Liberia "recounted how immunization teams had worn personal protective equipment and that this caused communities to be afraid" (p. 86) [22] of polio immunization.

Related to the beliefs of individuals, one study highlighted fear of contracting the disease as a facilitator to polio immunization uptake. The authors of one study in Nigeria stated that participants felt that "their fear of disease overshadowed the perceived risks of vaccination" (p. 3324) [15].

**Community trust.** Community trust in organizations and governments was also a key theme in the implementation of polio programs. Distrust stemming from political and social exclusion hindered programs' credibility in certain communities. On the other hand, trust served as a facilitator in program uptake in certain settings where trust in government services was high. Under this theme, two studies identified three barriers. One study stated this theme as a facilitator to polio immunization acceptance.

One study explicitly identified government distrust as a barrier to intervention acceptance in India [18]. The respondents of this study "were more likely to blame the government and polio workers" when discussing reasons for avoiding vaccination (p. 14) [18]. In addition, one study mentioned the resistance of participants living in Pakistan due to the lack of basic health facilities in their communities [20]. This lack of infrastructure was a primary cause of mistrust, and, consequently, resistance to OPV. Moreover, this study also discussed refusal of intervention uptake as a means of protest against the government in order to address their political and economic problems.

One study described increased trust in government health services and thus acceptance of vaccination when improvements were made to these services across seven countries (Nepal, India, Pakistan, Ethiopia, Nigeria, Rwanda, and Angola) [23]. A frontline worker from this study stated that "one manifestation of this trust [in government health services] was acceptance of vaccination" (p. 15) [23].

**Infrastructure.** Infrastructure was another theme identified in the implementation of polio programs. General infrastructural issues impeded effective implementation and limited efforts to reach vulnerable populations. Conversely, in the presence of good infrastructure and resource mobilization, the implementation of polio programs was deemed successful. Under this theme, nine studies mentioned a total of five barriers, and three studies mentioned a total of three facilitators.

One study highlighted that in Nigeria, the general lack of basic necessities, including drug availability in hospitals and water service, is a barrier to future immunization uptake [15].

Eight studies highlighted available resources as a challenge to the effective implementation of polio programs [14, 17, 18, 21, 23–26]. The lack of access to healthcare was mentioned by one study in Niger, where the majority of respondents noted inaccessibility as impeding routine immunizations and acute flaccid paralysis surveillance [24]. Two additional studies highlighted financial, organizational, operational, logistical, and quality barriers to the delivery of immunization services across several countries, including Nepal, India, Pakistan, Ethiopia, Nigeria, Rwanda, and Angola [14, 23]. The study conducted in Ethiopia listed "shortage of vaccine and supplies, non-functionality of refrigerators, lack of training, cancellation of immunization sessions, and unavailability of health posts to deliver the services" (p. 33) as possible reasons for low immunization [14]. Moreover, the unavailability of eligible children was considered a barrier by four studies across Pakistan, Afghanistan, and India [17, 19, 25, 27]. Children were frequently missed during polio vaccination visits and were not followed up with at a later time. In a 2017 study in Pakistan, "38 percent of children were not vaccinated with OPV," and the most frequent reason given was "the unavailability of children at home during polio vaccinator visits" (p. 29) [19]. Lack of general infrastructure, including access to hard-to-reach flood plains, weak health infrastructures, and insufficient financing for strategy implementation, impeded effective polio implementation in one study [26].

Furthermore, three studies listed the availability of resources as a facilitator in some settings [25, 28, 29]. First, one study highlighted a good social mobilization system in Afghanistan, where mobilizers visited over "90 percent of targeted, low-performing communities in the South" (p. 170) prior to the program's implementation [25]. In addition, the finding and mapping of marginalized populations across seven countries in South Asia and sub-Saharan Africa facilitated the targeting of children who may have been missed during polio visits in one study [28]. Lastly, one study in Uganda highlighted increased accessibility to immunization due to close National Immunization Days (NID) posts as facilitating individuals' participation in routine immunization campaigns, citing "NID posts very near and within a short walking distance" from their homes (p. 367) [29].

**Beliefs about the intervention.** The most prominent theme discovered in the implementation of polio programs surrounded beliefs related to the intervention. Negative beliefs about the polio vaccine emerged from cultural beliefs, religious conceptions, and inadequate awareness of the vaccine's purpose and effectiveness. However, increased knowledge and positive beliefs about the vaccine's efficacy against disease facilitated program implementation in other settings. Under this theme, 11 barriers from 15 studies, and four facilitators from seven studies were identified.

Anti-immunization debates were thought to negatively contribute to the community's trust in the vaccine. For instance, anti-immunization debates challenged the immunization program in India [30]. The authors of this study highlight that frequent debates challenging immunization programs "began to influence the community's trust in the vaccines" (p. 7).

When considering the presence of particular beliefs about the intervention, 14 studies mentioned a total of nine barriers. An additional seven studies listed a total of four facilitators.

Propaganda and mistrust against vaccines and their efficacy were mentioned by two studies as the reasons for parental vaccine refusal in India [17, 31]. An additional study identified the importance of cultural beliefs in the decision to reject the vaccine in Niger [24]. Categorized under knowledge and beliefs about the intervention, seven studies mentioned refusal of polio program uptake due to a lack of information, ignorance, and illiteracy across Pakistan, Nigeria, India, and the United States of America [15–17, 19, 21, 27, 32]. For instance, in Nigeria, the majority of refusal respondents mentioned "drinking of clean water as a way to avoid polio virus, [while] others identified [that] ways to protect against polio included proper care and belief in God" (p. 3326) [15]. The belief that vaccination was unnecessary was cited as an

indicator of refusal and thus a barrier to intervention uptake across four studies [15, 21, 23, 25]. Moreover, a study in India identified vaccine-resistant areas in a particular district as a barrier [30]. Safety and religious misconceptions regarding the vaccine were mentioned by two studies, which included witchcraft accusations and Westernized medicines that threaten religious credentials [19, 23]. In Pakistan, the misconception that "the vaccine is not 'halal' or [that it is] impermissible in the Islamic Law" (p. 27) [19] was a major reason for vaccine refusal. The perception of age, particularly being "too old," and lax or disinterested parents were deemed as reasons by participants in one study to avoid polio vaccination in the United States of America [33]. Additionally, the reliance on religious leaders that oppose OPV uptake fueled skepticism and resistance in one study in Pakistan [20]. Lastly, one study mentioned mothers avoiding OPV by running away with their children as a barrier to polio campaigns in India [17].

Parents' moral obligation to vaccinate their children and their belief that they cannot be deviant was mentioned by one study as a facilitator to polio intervention use in India [31]. Additionally, four studies across the United States of America, Uganda, India, and Pakistan highlighted the belief in the vaccine's ability to protect against polio as a facilitator [20, 21, 29, 30]. Participants of one study in Uganda said that "the major aim [of the vaccine] was to 'weaken' the disease and/or 'strengthen' the children's capability in fighting diseases" (p. 367) [29]. The public's awareness of polio vaccines and the intervention in general were mentioned by three studies as a positive influence on uptake [19, 22, 23]. Lastly, one study described the decrease in religious resistance to OPV as associated with the defeat of militants and increased involvement of religious scholars in the polio vaccination campaigns in Pakistan [20].

**Influential opinions.** Influential opinions was a theme that emerged that was closely tied with acceptance of the polio program. Some religious leaders and health professionals were not in support of the polio programs. Because the opinions of these groups were heavily influential in parents' decision-making, they served as a hindrance to program implementation. On the other hand, religious leaders and health professionals who served as champions of the vaccine were helpful in facilitating program implementation. Under this theme, five barriers were identified by six studies, and three facilitators were identified by four studies.

The meaning and values attached to the intervention by health workers and opinion leaders was cited by two studies as a challenge in the implementation of polio interventions [29, 31]. First, one study in India stated medical practitioners' opposition against repeated rounds of polio vaccination as a barrier to implementation [31]. These private medical practitioners "advised their clients against vaccination as they thought it was unnecessary for children in Kerala" (p. 6) [31]. Moreover, another study conducted in Uganda highlighted the influence of health workers and opinion leaders, as laypeople emulated their refusal to vaccinate their own children [29]. In both cases, reservations about the intervention were influenced by individuals who may be involved at some capacity in the intervention or whose opinions concerning the intervention may be perceived as credible.

One study identified health workers and opinion leaders' refusal to vaccinate their children as a negative influence on participants in Uganda [29].

Protests by alternative medicine providers, including active discouragement by homeopathic practitioners, was a barrier mentioned by one study in India [30]. For instance, "many homeopathic practitioners have actively discouraged their clients from immunizing their children," exerting strong influence on households in northern Kerala and "convincing them against immunization" (p. 7). Another study highlighted the reliance on religious leaders that oppose OPV as a barrier to polio immunization interventions in Pakistan [20].

Four studies described champions as facilitators to intervention implementation. First, three studies highlighted the influence from health-focused champions in endorsing the

uptake of polio vaccination, thus positively influencing the decision-making process of participants across the United States of America, India, Nepal, Pakistan, Ethiopia, Nigeria, Rwanda, and Angola [21, 23, 30]. Moreover, one study described the involvement of religious scholars as decreasing religious resistance to OPV and increasing the acceptance of the vaccine [20]. Participants of this study noted that "the involvement of religious scholars in the OPV campaigns also helped the cause of polio eradication," decreasing the number of non-compliant parents (p. 3701). Lastly, the decrease in religious resistance to OPV through the defeat of militants and the increased involvement of religious scholars positively influenced polio vaccination uptake in one study [20].

**Intervention design.**   Intervention design was another theme that emerged from the implementation of polio programs. Strategies to reach target populations were a challenge as the resources needed for such mobilization were not available or convenient in some settings. However, in other settings, strategies including policies that enforce polio vaccination served as facilitators to program implementation. Under this theme, two barriers and four facilitators were mentioned across three and four studies, respectively.

Four articles listed aspects of design quality and packaging as either a barrier or facilitator to polio immunization programs. First, the absence or failure of vaccination teams to visit homes to administer vaccines to children in Afghanistan and Pakistan were listed as barriers by two studies [19, 25]. The study in Pakistan revealed that "29 percent of children did not receive OPV due to absence of polio vaccination teams from their duty" [19] (p. 29), underscoring the gap in implementation programs to reach vulnerable children. Moreover, the frequency of visits needed for booster shots of the polio vaccine in Pakistan was mentioned by one study as a deterrent to vaccination uptake [16].

On the other hand, the multiple polio campaigns in numerous countries, including Nepal, India, Pakistan, Ethiopia, Nigeria, Rwanda, and Angola, were deemed a facilitator to implementation; one study revealed that many respondents asserted the effectiveness of several campaigns [23]. In a 2016 study, respondents stated that "people were happy with polio campaigns, largely because they no longer had to fear polio" (p. 9) [23].

The Pakistani government mandating polio vaccination in order to fly abroad was underscored as a facilitator to intervention uptake by one study [20]. By mandating "presenting polio vaccination certificates to immigration officers at airports before flying abroad" (p. 3697), the government was able to enforce legal measures to encourage the uptake of vaccines.

Moreover, two studies mentioned coercive means as a facilitator to intervention uptake [15, 31]. In one study, emphasis on coverage results and the use of coercive means restricted options for refusal in India [31]. Moreover, another study in Nigeria noted that "a majority of the acceptors presented their children for vaccination because they had been told to do so" (p. 3325) by health workers [15]. The way in which the intervention was presented in these two sites (deeming it to be mandatory) facilitated uptake.

**Geo-politics.**   Geo-politics was also a theme that emerged from the implementation of polio programs. This theme emerged from several barriers that could not be coded using the CFIR. These barriers were external to the intervention itself but negatively influenced the implementation of programs across several settings as many children who were eligible for vaccinations were inaccessible. These barriers included the relocation of mothers mentioned by two authors [14, 17], the inaccessibility to reach children highlighted by one author [25], and conflict and security issues described by two studies conducted in Pakistan [16, 34]. For example, one author noted that "the global war on terror and the geopolitical situation in Afghanistan impacted polio eradication and other immunization efforts in the country" (p. 7) [34].

## Discussion

This systematic review identified multiple barriers and facilitators to the implementation of polio vaccination and eradication programs from a global perspective across 20 published articles between 1960 and 2019. Our analysis was dominated by India, Nigeria, Pakistan, Ethiopia, and Afghanistan. The barriers and facilitators identified in this study are consistent with the literature published in the last two decades [2, 8]. This systematic review has implications for current and future vaccination programs.

A key lesson from this study is that implementation practitioners should recognize the relevance of religious beliefs and leaders and the need to include them in public awareness campaigns on the need to vaccinate. This implication is related to the themes of community trust, beliefs about interventions, and fear. Beliefs related to interventions constituted a crucial factor in implementing polio vaccination and eradication programs as both barrier and facilitator. The beliefs fell into two categories: (1) geo-political and (2) vaccine efficacy concerns and misconceptions and religious concerns about polio interventions. Some people doubted the true motive of the vaccination. Coupled with their distrust of the government, they refrained from the intervention, highlighting the need to undertake community engagement, public awareness campaigns, education strategies, and trust-building among the populace in establishing and increasing uptake [8]. Examples of these strategies have been identified across several studies, including the positive reception of good social mobilization systems in Afghanistan and the increase in public trust and knowledge through multiple polio campaigns in Nepal, India, Pakistan, Ethiopia, Nigeria, Rwanda, and Angola [23, 25]. The findings of these studies reflect the lessons learned in newer studies, including the significance and effectiveness of good social mobilization strategies for engaging communities in polio vaccination campaigns in India, Pakistan, and across various remote settings in Africa and Asia [35–38]. These strategies will open avenues to highlight the efficacy and relevance of the vaccine to communities and provide answers to questions that bother them. Cultural and religious misconceptions were critical, especially relating to Islamic beliefs and some Muslims who believed that the vaccine is not halal [19]. These findings concur with Mshelia et al., who found competing belief systems among caregivers founded on Islamic principles and misconceptions as major factors in polio vaccination decision-making [2]. Ataullahjan et al also found that some Muslims in Pakistan believed that some haram (impermissible) products were included in the vaccine [8]. These religious beliefs are often misconceptions and not scientifically supported, emphasizing the need to engage Imams and other religious leaders in addressing polio vaccine misconceptions.

Additionally, this review revealed that elements of health systems, including infrastructure and human resources, were still relevant in implementing polio vaccination and eradication programs. Lack of access to general and health infrastructure, including healthcare, health facilities, and financial, organizational, and logistical constraints, served as key barriers to the vaccination interventions, highlighting the need for implementers to take health system capacity into account when considering vulnerable groups at the design stage of interventions [14, 18, 20, 23, 24, 26]. Similarly, newer studies support these findings, as they attempt to highlight the need for surveillance capacity and health service delivery across many contexts, including: remote and conflict-affected areas [39–41]. For instance, common implementation barriers in regions of the Democratic Republic of Congo and Ethiopia include inaccessibility issues caused by gaps in human resources, supply chain, and finance [39]. In this study, the strategies that best addressed these barriers included adapting service delivery approaches, investing in health systems capacity, and strengthening accountability and planning [39]. The availability of human resources in terms of adequate staff and technical support is also critical in the

implementation of polio vaccination and coverage. In Afghanistan, for instance, vaccination teams were unable to reach between 7.6 to 56.6 percent of eligible children due to the absence of adequate staff [25]. Conversely, adequate human resources for health in Rwanda enabled staff to find, map, and repeatedly visit vaccine-eligible populations that were not reached by other health services [28]. A similar study in East China revealed the strong positive association between the density of vaccination workers and immunization coverage in the Zhejiang province [42]. In this study, a higher density of vaccination workers improved the availability of vaccination services, including: the administration of vaccinations, the education of parents on immunization, raising general awareness of health in the community, and ensuring the continuous vaccine supply [42]. Human resources are essential in vaccination programs; hence, thinking of new ways to incentivize and motivate staff to visit rural areas and vulnerable groups may be critical for effective implementation.

Reliance on the opinions and recommendations of religious leaders/scholars, health professionals, and health-focused champions played an important role in implementing polio vaccination and eradication programs, highlighting the need to research ways to motivate people to partake in polio immunization programs. Encouraging influential individuals within a particular context to share their opinions, thoughts, and attitudes toward a particular health intervention during the planning stages may increase knowledge sharing and possible collaboration to enhance community sensitization. Through the support and championing of these influential individuals, health interventions may lead to increased receptiveness by the public. Several studies in this review found that uneducated people went to religious leaders and alternative health providers as their main source of health information, demonstrating over-reliance on these individuals as opposed to recommendations set out by government bodies and workers affiliated with national systems (including health professionals) [20, 30]. Polio vaccination interventions could take advantage of the trust in religious and traditional leaders by educating these leaders on vaccines and the need to increase coverage [20, 30]. In Pakistan, women and religious leaders have played a central role in addressing vaccine misconceptions related to religious concerns [43]. First, strategies to increase trust include relying on the female workforce of vaccinators within local communities who have already fostered trust [43]. Second, to increase vaccine-related knowledge and to dispel religious misinformation, Muslim religious leaders worked closely with the national government to endorse pro-vaccine fatwas (a formal ruling on a point of Islamic law given by a recognized authority), as well as to deliver pro-vaccine sermons and establish a 24/7 WhatsApp polio help line and call center [43, 44]. Religious and traditional leaders are important in vaccination programs, so research on various methods of community engagement and leader inclusion in polio interventions will be relevant to implementers. Additionally, future research could provide important insights by considering the perspectives of traditional and religious leaders with regards to vaccination.

This study found that the design of the intervention in terms of frequency of visits and campaigns was relevant in implementing polio vaccination and eradication programs. Mshelia et al. found that revisits created suspicion in parents' minds and impacted the program negatively [2]. Meanwhile, it was found that multiple campaigns were a facilitator to the program as they might address fears and misconceptions associated with vaccination [23]. This result highlights the need to address suspicion in parents' minds in areas where myriad visits are undertaken. The quality of a polio intervention design is a major factor in vaccine uptake and should be prioritized by implementers.

Finally, there were factors beyond the intervention design, geo-political factors, that influenced implementation and are unique to low- and middle-income countries, rendering them difficult to code within the CFIR constructs available. Implementation science theories, models, and frameworks are typically applied to the analysis of high-income countries, which often

results in different implementation determinants and therefore the discovery of new influences beyond the contents included in the CFIR, for instance [45]. One study described the pragmatic adaptations that were made to the CFIR to clarify the relationships between determinants and outcomes [46]. In this study, adaptations to broad categories of outcomes were made and informed by the Reach, Effectiveness, Adoption, Implementation, and Maintenance (RE-AIM) framework and the Implementation Outcomes Framework (IOF), which consider both implementation and innovation outcomes [46]. In our review, geopolitical factors were critical to implementation but were not represented in the existing CFIR framework. This finding implies that important contextual factors may not be accounted for in implementation frameworks that intend to highlight the health system environment and inform implementation and evaluation strategies. Because these geopolitical factors were challenges faced in numerous settings, a new proposed domain may be more applicable to similar contexts and may be more appropriate in informing implementation [45, 46].

## Strengths and limitations

This review employed the multi-context approach to synthesize data, providing a wide range of evidence that allows cross-setting comparison and applying study findings to broader settings [47, 48]. Although such an approach offers transferable patterns of findings at a global level, they may be too general to target important local characteristics or audiences, leading to the study overlooking the relevance of specific contexts and blurring critical differences across different studies [47, 48]. It might also appear that the study is ignoring the significance of context in evidence synthesis [48]. Nonetheless, barriers and facilitators associated with polio vaccination and eradication are not too varied. Thus, contextual heterogeneity is mostly minimal in such programs aside from outliers such as political unrest that impede vaccination in certain parts of the world.

This study also presents several limitations that should be taken into consideration when interpreting the results. For instance, we limited the inclusion criteria to primary research articles, and as such, valuable information from secondary sources might have been overlooked. In addition, the selection of a determinant framework (CFIR) is a potential limitation as such frameworks have been criticized for their inadequacy in addressing causal mechanisms or how change takes place [49]. However, the use of the CFIR allowed the findings to be placed in the context of the wider implementation research literature and has been successfully applied in similar systematic reviews [50, 51]. Finally, we extracted only qualitative-oriented data from each eligible article; thus, other significant results might have been overlooked.

## Conclusion

This study identifies a wide range of barriers and facilitators to polio vaccination implementation across the globe, adding to the scarce body of literature on barriers and facilitators from an implementation perspective and using a determinant framework. Seven themes emerged from this review: fear, community trust, infrastructure, beliefs about the intervention, influential opinions, intervention design, and geo-politics. The diversity of factors among different groups of people or countries highlights the relevance of contexts within which polio eradication programs boost intervention coverage and capacity, and provides policymakers, practitioners, and researchers with a tool for planning and designing polio immunization programs.

## Supporting information

**S1 Data. PRISMA checklist.**
(DOCX)

**S2 Data. Search strings.**
(DOCX)

**S3 Data. MMAT/CASP assessment.**
(DOCX)

**S1 Table. Barriers to polio vaccination and eradication programs.**
(DOCX)

**S2 Table. Facilitators to polio vaccination and eradication programs.**
(DOCX)

**S3 Table. Study characteristics.**
(DOCX)

## Author Contributions

**Conceptualization:** Obidimma Ezezika.

**Data curation:** Obidimma Ezezika, Aiman Farheen, Anuradha Chauhan, Kathryn Barrett.

**Formal analysis:** Meron Mengistu, Eric Opoku, Aiman Farheen, Anuradha Chauhan.

**Investigation:** Kathryn Barrett.

**Methodology:** Obidimma Ezezika, Meron Mengistu, Aiman Farheen, Anuradha Chauhan, Kathryn Barrett.

**Project administration:** Obidimma Ezezika.

**Resources:** Obidimma Ezezika.

**Supervision:** Obidimma Ezezika, Kathryn Barrett.

**Validation:** Meron Mengistu, Eric Opoku, Kathryn Barrett.

**Writing – original draft:** Meron Mengistu, Eric Opoku, Kathryn Barrett.

**Writing – review & editing:** Obidimma Ezezika, Meron Mengistu, Eric Opoku, Kathryn Barrett.

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
