## [Decision Letter · Decision Letter 0]

16 Aug 2022

PGPH-D-22-00988

Barriers and Facilitators to Polio Vaccination and Eradication Programs: A Systematic Review

Dear Dr. Ezezika,

Thank you for submitting your manuscript to PLOS Global Public Health. After careful consideration, we feel that it has merit but does not fully meet PLOS Global Public Health’s publication criteria as it currently stands. Therefore, we invite you to submit a revised version of the manuscript that addresses the points raised during the review process.

We look forward to receiving your revised manuscript.

Kind regards,

Ejemai Eboreime, MD, MSc, PhD

Academic Editor

Journal Requirements:

1. Thank you for your submission at PLOS ONE. In order to meet journal requirements for reporting and reproducibility for Systematic Reviews, at this time we request that you please update the Methods section to report the search key words (strings) used to collect your data in sufficient detail for another researcher to replicate the search.

3. Please provide separate figure files in .tif or .eps format and remove the embedded figure from the manuscript file.

Additional Editor Comments (if provided):

Reviewers' comments:

Reviewer's Responses to Questions

**Comments to the Author**

1. Does this manuscript meet PLOS Global Public Health’s publication criteria? Is the manuscript technically sound, and do the data support the conclusions? The manuscript must describe methodologically and ethically rigorous research with conclusions that are appropriately drawn based on the data presented.

Reviewer #1: Yes

Reviewer #2: Yes

2. Has the statistical analysis been performed appropriately and rigorously?

Reviewer #1: N/A

Reviewer #2: Yes

3. Have the authors made all data underlying the findings in their manuscript fully available (please refer to the Data Availability Statement at the start of the manuscript PDF file)?

Reviewer #1: Yes

Reviewer #2: Yes

4. Is the manuscript presented in an intelligible fashion and written in standard English?

Reviewer #1: Yes

Reviewer #2: Yes

5. Review Comments to the Author

Reviewer #1: This research was done using appropriate methodological techniques and rigor according to recognised criteria and templates for qualitative reviews. The analysis of the qualitative findings was done excellently well using the CFIR framework and as such, statistical analysis was not applicable. It was well written not requiring further editing.

It was interesting to see this paper, a more global multi-context review aside the available reviews limited to a region (e.g SSA). This provided a richer evidence base and data though some of the findings were not entirely new because the challenge with Polio in recent years has been limited to certain LMIC which the review still captured. I was particularly happy to see the health system of Rwanda in the discussion.

I was also happy to see the CFIR approach used to analyse the findings. However, there was no statement on the comparison of this approach with the best-fit framework which I had looked forward to reading. It will ne good to make a methodological statement on this bearing in mind that previous studies that can be compared with were done in different times, restricted to certain regions.

It will be good for the authors to add a section on the strengths of this review in addition to the limitations. In reviewing these two sections (strengths and limitations), take into account the ongoing discussions of how multi-context reviews should be used compared to single-context reviews. This is critical and will provide insights into how the findings should be used or utilised by policymakers and practitioners. Moreso, your review was global, highly heterogenous in nature comprising both high income countries and LMIC. kindly refer to this for more information (Booth A, Mshelia S, Analo CV, Nyakang'o SB. Qualitative evidence syntheses: Assessing the relative contributions of multi‐context and single‐context reviews. Journal of Advanced Nursing. 2019 Dec;75(12):3812-22 ) assessed here https://doi.org/10.1111/jan.14186.

Welldone to the authors for a good job.

Reviewer #2: Reviewer Comments

This manuscript studied Barriers and Facilitators to Polio Vaccination and Eradication Programs: A Systematic Review. The paper is very interesting, but there are some minor issues. However, some other problems in the manuscript are still concerned in the following:

1)More keywords are suggested.

2)Divide your abstract in different sections (i.e Background, objective, methods, results, conclusion and recommendations)

3)A section of study area and data is suggested.

4)The language should be polished.

5)Authors use old references sometimes in the paper(eg reference number 3,4,5,12 and 15 etc). Please use the updated (2020-21-22) study or research.

6. PLOS authors have the option to publish the peer review history of their article (what does this mean?). If published, this will include your full peer review and any attached files.

**Do you want your identity to be public for this peer review?** For information about this choice, including consent withdrawal, please see our Privacy Policy.

Reviewer #1: **Yes: **Suleiman Mshelia

Reviewer #2: No

---

## [Editor Report · Decision Letter 1]

21 Sep 2022

PGPH-D-22-00988R1

What are the Barriers and Facilitators to Polio Vaccination and Eradication Programs? A Systematic Review

Dear Dr. Ezezika,

Thank you for submitting your manuscript to PLOS Global Public Health. After careful consideration, we feel that it has merit but does not fully meet PLOS Global Public Health’s publication criteria as it currently stands. Therefore, we invite you to submit a revised version of the manuscript that addresses the points raised during the review process.

We look forward to receiving your revised manuscript.

Kind regards,

Ejemai Eboreime, MD, MSc, PhD

Academic Editor

Journal Requirements:

1. Please update the Methods section to report the search key words (strings) used to collect your data in sufficient detail for another researcher to replicate the search.
---

## [Editor Report · Decision Letter 2]

25 Oct 2022

What are the Barriers and Facilitators to Polio Vaccination and Eradication Programs? A Systematic Review

PGPH-D-22-00988R2

Dear Dr. Ezezika,

We are pleased to inform you that your manuscript 'What are the Barriers and Facilitators to Polio Vaccination and Eradication Programs? A Systematic Review' has been provisionally accepted for publication in PLOS Global Public Health.

Best regards,

Ejemai Eboreime, MD, MSc, PhD

Academic Editor